# Hypoxia-Induced Adaptations of Embryonic Fibroblasts: Implications for Developmental Processes

**DOI:** 10.3390/biology13080598

**Published:** 2024-08-08

**Authors:** Zeyu Li, Delong Han, Zhenchi Li, Lingjie Luo

**Affiliations:** 1College of Pharmaceutical Sciences, Yunnan University of Chinese Medicine, Kunming 650500, China; l13703564582@163.com; 2Marshall Laboratory of Biomedical Engineering, Institute for Inheritance-Based Innovation of Chinese Medicine, Shenzhen University Medical School, Shenzhen 518055, China; 17883686338@163.com (D.H.);

**Keywords:** hypoxia, HIF1a, CoCl_2_, MEFs, embryonic development

## Abstract

**Simple Summary:**

For centuries, scientists have studied how a single cell develops into a complex organism. During this intricate process, any error can cause birth defects, some even fatal. These defects can affect the brain, heart, face, limbs, and multiple tissues. Understanding how these defects occur is crucial for developmental defects. Mammalian embryonic development occurs under hypoxia. This low-oxygen environment helps form new blood vessels, organs, and tissues. Embryonic fibroblasts are special cells that play a vital role in building tissues and organs during development. Understanding how embryonic fibroblasts work under hypoxia could lead to new treatments for birth defects and other developmental problems. Our study shows that hypoxia can cause several changes in embryonic fibroblasts, including increased migration, metabolic changes, the production of ROS, and apoptosis. These changes are triggered by the activation of various pathways and genes, including HIF1a. We also identified new genes that are regulated by hypoxia. These findings highlight the importance of low oxygen in regulating the functions of embryonic fibroblasts, and further research is needed to understand the mechanisms involved. This knowledge could lead to new treatments for developmental disorders and tissue regeneration.

**Abstract:**

Animal embryonic development occurs under hypoxia, which can promote various developmental processes. Embryonic fibroblasts, which can differentiate into bone and cartilage and secrete various members of the collagen protein family, play essential roles in the formation of embryonic connective tissues and basement membranes. However, the adaptations of embryonic fibroblasts under hypoxia remain poorly understood. In this study, we investigated the effects of hypoxia on mouse embryonic fibroblasts (MEFs). We found that hypoxia can induce migration, promote metabolic reprogramming, induce the production of ROS and apoptosis, and trigger the activation of multiple signaling pathways of MEFs. Additionally, we identified several hypoxia-inducible genes, including *Proser2*, *Bean1*, *Dpf1*, *Rnf128*, and *Fam71f1*, which are regulated by HIF1α. Furthermore, we demonstrated that CoCl_2_ partially mimics the effects of low oxygen on MEFs. However, we found that the mechanisms underlying the production of ROS and apoptosis differ between hypoxia and CoCl_2_ treatment. These findings provide insights into the complex interplay between hypoxia, fibroblasts, and embryonic developmental processes.

## 1. Introduction

Hypoxia is often considered a pathological phenomenon, associated with multiple diseases, such as pulmonary hypertension [1], rheumatoid arthritis (RA) [2], inflammatory bowel disease [3] and cancer [4]. However, mammalian embryonic development occurs in a low-oxygen environment. Oxygen concentrations range from 1% to 5% in the uterine environment [5]. In the mouse embryo, cells with hypoxia are widespread until the maternal and fetal blood interface is established around mid-gestation [6].

Hypoxia has several important effects on embryonic development. It is associated with the development of new blood vessels (angiogenesis) in embryos of *Japanese quail* [7]. Hypoxia also plays a role in the development of specific organs and tissues, such as the heart [8], brain [9], and lung [10]. Furthermore, hypoxia stimulates the production of erythropoiesis [11] and increases the expression of genes involved in cell survival and proliferation. Overall, hypoxia is a complex and dynamic process with significant implications for embryonic development. Understanding its role is critical for developing new therapies for birth defects and other developmental disorders.

Embryonic fibroblasts play crucial roles in embryonic development. They can differentiate into bone and cartilage and secrete various members of the collagen protein family, which play essential roles in the formation of embryonic connective tissues and basement membranes [12]. In addition, embryonic fibroblasts participate in processes such as cell migration, proliferation, and signaling during embryogenesis. They interact with other types of cells to collectively promote the normal development of the embryo. Therefore, the proper function of embryonic fibroblasts is essential for the correct progression of embryonic development. Overall, embryonic fibroblasts are essential cells in embryonic development, playing diverse roles in tissue formation, angiogenesis, and immune regulation.

Embryonic fibroblasts typically appear in the primitive mesoderm tissue of the embryo, which occurs around the fourth to fifth day of embryo development, specifically during the late blastocyst stage to the early implantation period [13]. Thus, embryonic fibroblasts are maintained in a low-oxygen environment during early embryonic development. Mouse embryonic fibroblasts (MEFs) exhibit significant variation and some can be induced to differentiate into cell types of fat, muscle, and bone lineages [14]. Although there has been extensive research on the relationship between embryonic development and hypoxia, the specific functions of embryonic fibroblasts under hypoxia remain unclear. Thus, understanding the behavior and functions of embryonic fibroblasts under hypoxia is crucial.

CoCl_2_ is an iron chelator that functions by mimicking the effects of hypoxia [15]. It achieves this by replacing Fe^2+^ in the oxygen-sensing heme group of the hypoxia-inducible factor (HIF) prolyl hydroxylase (PHD) enzyme. This replacement inhibits PHD activity, preventing the degradation of HIF-1α, a key regulator of the cellular response to low oxygen levels [16]. Consequently, HIF1α accumulates, leading to the expression of genes associated with hypoxic adaptation, including those involved in angiogenesis, glycolysis, and cell survival [17]. CoCl_2_ is widely used to mimic hypoxic conditions in a variety of cell types, such as HepG2 [18], MCF-7 cells [19], and colorectal cancer cells [20].

Here, we showed that hypoxia can induce migration, promote metabolic reprogramming, and induce the production of reactive oxygen species (ROS) and apoptosis of MEFs. Hypoxia can induce multiple pathways in MEFs, leading to the expression of genes regulated by multiple TFs, including p53, SP1, NF-κB, STAT3 and HIF1a. In addition, we identified several new genes, including *Proser2*, *Bean1*, *Dpf1*, *Rnf128*, and *Fam71f1*, regulated by HIF1a. Furthermore, we found that CoCl_2_ treatment can induce similar effects on MEFs, including migration, metabolic reprogramming, ROS production, apoptosis, and the activation of HIF signaling pathways. However, our study also revealed differences in the mechanisms underlying these effects between hypoxia and CoCl_2_ treatment. Our findings highlight the importance of hypoxia in regulating the functions of MEFs, and underscore the need for further research to elucidate the molecular mechanisms underlying these adaptations. Understanding the role of hypoxia and MEFs responses in embryonic development could lead to novel therapeutic strategies for developmental disorders and tissue regeneration.

## 2. Materials and Methods

### 2.1. Reagents

CoCl_2_ (Macklin, Shanghai, China, cat. no.C15576355), crystal violet (Solarbio, Beijing, China, cat. no. C8470) and HIF-1α-IN-2 (MCE, cat. no. HY-115903).

### 2.2. Cell Culture

BALB/3T3 clone A31 cells, sourced from Wuhan Pricella (Wuhan, China), maintained in DMEM enhanced by 10% FBS and 100 U/mL P/S, under 37 °C and 5% CO_2_ ambiance.

### 2.3. ROS Detection

Cultured in 24-well plates, cells underwent overnight incubation before being treated with CoCl_2_ for 24 h. Subsequently, cells were incubated with 10 µM DCFH-DA (Beyotime, cat. no. S0033S) in serum-free DMEM for 20 min. We washed the cells three times with serum-free DMEM. 2′,7′-dichlorofluorescein (DCF) was measured under Cytation1 (Bio-Tek; Winooski, VT, USA) with spectra of 469 nm excitation/525 nm emission. The fluorescence intensity was measured by Image J. Briefly, images were opened in ImageJ 8 software. Background subtraction was performed using the rolling ball algorithm to remove non-specific signals. The intensity of the ROS signal was measured. All ROS intensity values were normalized to the intensity of the control group to account for differences in dye concentration.

### 2.4. Cell Viability Assay

Cell viability was measured with a CCK-8 assay kit (Meilunbio^®^, Dalian, China, cat. no. MA0218) following instructions. BALB/3T3 clone A31 cells were cultured in a 96-well plate. Following overnight incubation, cells were subjected to serum starvation for 6 h before treatment with CoCl_2_ for 48 h. DMSO served as the control. The working solution, prepared by mixing DMEM without FBS with the CCK-8 solution at a ratio of 9:1, was added (100 μL) to each well and incubated for 30 min. Absorbance readings were taken at 450 nm using the microplate reader (Bio-Tek; Winooski, VT, USA).

### 2.5. Wound Healing Assay

After seeding 20,000 cells into 96-well plates and allowing for overnight incubation, BALB/3T3 clone A31 cells were serum-starved for 6 h before a scratch assay was performed. Subsequent imaging of the wounded area was performed using a microplate reader (Bio-Tek; Winooski, VT, USA). The migration of cells into the scratched zone was assessed at the onset (0 h) and after 24 h. Wound healing was quantified by the formula: % scratch = 100 × (width at T0 − width at T24)/width at T0. Data analysis was conducted using the Bio-Tek Cytation1 and Prism 8 software (San Diego, CA, USA).

### 2.6. Western Blot Analysis

For protein extraction, cells were rinsed with pre-cooled PBS and subsequently lysed employing RIPA buffer (Beyotime, Nantong, China, cat. no. P0013B), added with proteinase (MIKX, Shenzhen, China, cat. no. DB612A) and phosphatase inhibitors (MIKX, Shenzhen, China, cat. no. DB615). Following scraping and collection, the resultant lysate solutions were subjected to centrifugation at 12,000 rpm for 15 min at 4 °C. Subsequently, equal amounts of protein were separated on a 10% SDS polyacrylamide gel and transferred onto a PVDF membrane (Millipore, Billerica, MA, USA); then, the membranes were blocked in 5% non-fat dry milk in TBST for a duration of 1 h at room temperature and appropriately diluted with primary antibodies in 3% BSA overnight at 4 °C. β-actin (Immunoway, cat. no. YM-3028) was acquired from Immunoway (Plano, TX, USA), along with HIF-1α (NOVUS, Saint Charles, MO, USA, cat. no. D108267-4). The subsequent step involved an incubation period with secondary antibodies, diluted in 3% BSA, for a duration of 2 h at room temperature. The membranes were then subjected to development for chemiluminescence detection, utilizing an ECL detection kit (MIKX, cat. no. MK-S700). Subsequently, the intensities of the bands were quantified using ImageJ 8 software.

### 2.7. Quantitative Real-Time PCR Analysis

RNA was extracted using RNAiso Plus (Takara, Kyoto, Japan, cat. no. 9108), reverse transcribed to cDNA, and amplified by PCR (Vazyme, Nanjing, China, cat. no. R323). The quantitative real-time PCR utilized SYBR-Green PCR Master mix (MIKX, cat. no. MKG800-01). The primer sequences used are detailed in Table 1. Expression levels were calibrated against Actin values, and relative quantification was performed using the comparative Ct method. Data analysis was conducted using Prism 8 software.

### 2.8. In Vitro Migration Assay

We assessed cell migration using a Corning Transwell chamber (Corning, New York, NY, USA, cat. no. 3422). After seeding 100,000 cells into the Corning Transwell chamber we treated them with CoCl_2_ for 12 h. We fixed and stained the cells using 1% crystal violet, subsequently washing and drying them before imaging. The cell numbers were measured using ImageJ 8 software.

### 2.9. Procedure for Detection Apoptotic Cells Using Annexin V-FITC Kit

Cultured in 6-well plates, the cells underwent overnight incubation before being treated to CoCl_2_ for 24 h. Utilizing the annexin V-FITC kit (Meilunbio^®^, cat. no. MA0220) combined with PI staining, we detected cell apoptosis through flow cytometry.

### 2.10. The RNA-Sequencing and Data Analysis 

The RNA extraction and RNA sequencing (RNA-seq) methods used in this study are identical to those described in a previous publication [21]. Briefly, RNA integrity was assessed using the RNA Nano 6000 Assay Kit on the Bioanalyzer 2100 system (Agilent Technologies, Santa Clara, CA, USA) [22]. mRNA was subsequently purified, followed by first-strand cDNA synthesis. Second-strand cDNA synthesis was then performed using DNA Polymerase I and RNase H [23]. After adenylation of the 3’ ends of DNA fragments, adaptors with hairpin loop structures were ligated to prepare for hybridization. The library fragments were purified using the AMPure XP system [24]. PCR was then performed with Phusion High-Fidelity DNA polymerase, universal PCR primers, and index (X) primers. Finally, PCR products were purified (AMPure XP system, Brea, CA, USA) and library quality was assessed on the Agilent Bioanalyzer 2100 system. The clustering of the index-coded samples was performed on a cBot Cluster Generation System using the TruSeq PE Cluster Kit v3-cBot-HS (Illumina San Diego, CA, USA) according to the manufacturer’s instructions [25]. After cluster generation, the library preparations were sequenced on an Illumina NovaSeq platform, generating 150 bp paired-end reads.

Raw data (fastq format) were processed using the fastp (v0.23.2) software. Clean data (clean reads) were obtained by removing reads containing adapters, reads containing poly-N sequences, and low-quality reads. The reference genome index was built using Hisat2 v2.0.5, and paired-end clean reads were aligned to the reference genome using Hisat2 v2.0.5 [26]. The mapped reads of each sample were assembled by StringTie (v1.3.3b) using a reference-based approach [27]. Feature Counts v1.5.0-p3 was used to count the reads mapped to each gene [28]. FPKM for each gene was then calculated based on the gene length and the number of reads mapped to that gene. Differential expression analyses [29], GO and KEGG enrichment analyses of differentially expressed genes were performed using the cluster Profiler R package, with gene length bias correction [30,31]. GO terms with a corrected *p*-value less than 0.05 were considered significantly enriched by differentially expressed genes. The raw data from our RNA-sequencing experiments have been deposited in the GEO database (accession number: PRJNA1134336).

### 2.11. The Statistical Analysis

The results are expressed as the means ± SEM from at least three independent experiments. Data normality and homogeneity of variance were assessed using the Shapiro–Wilk. Two-group comparisons employed *t*-tests for equal variances and normal distributions, Wilcoxon tests for equal variances and non-normal distributions, and Welch’s *t*-tests for unequal variances and normal distributions. For more than two independent groups, one-way ANOVAs with Tukey’s post hoc tests were used for equal variances and normal distributions, while Kruskal–Wallis tests with Dunn’s post hoc tests were employed for equal variances and non-normal distributions. Welch’s ANOVAs with Games–Howell post hoc tests were applied for unequal variances and normal distributions. A significance level of *p* < 0.05 was adopted for all analyses. All statistical procedures were conducted using Prism 8 (San Diego, CA, USA).

## 3. Results

### 3.1. Establishing Hypoxic Models in Mouse Embryo Fibroblasts (MEFs) 

To investigate the potential links between hypoxia and MEFs, we established hypoxia- and CoCl_2_-induced MEF models. Compared with normoxia, the expression levels of HIF1α-regulated genes *Pdk1*, *Ldha*, *Glut1* and *Vegf* were significantly increased in MEFs under both hypoxia and CoCl_2_ treatment at different time points (Figure 1A,B). In addition, we found that hypoxia and CoCl_2_ treatment could promote the migration and invasion of MEFs (Figure 1C–J). Furthermore, both hypoxia and CoCl_2_ treatment could induce the production of ROS and apoptosis (Figure 1K–R), and inhibit the proliferation of MEFs (Appendix A).

### 3.2. Transcriptomic Profiles of Normoxic and Hypoxic in MEFs Reveal Common and Divergent Patterns 

To explore the detailed differences between hypoxia and normoxia in MEFs, we used mRNA-Seq to analyze the transcriptomes of MEFs under normoxia (*n* = 2) and hypoxia (*n* = 2, the hypoxic state was induced by exposing MEFs to a controlled environment with a reduced oxygen concentration of 1% for 24 h). A volcano plot created from the data reveals that 1348 genes were significantly upregulated in the hypoxia group compared with the normoxia group, while 1133 genes were significantly downregulated (Figure 2A). A heatmap was generated and showed significant differences in gene expression profiles between the hypoxia and normoxia groups (Figure 2B). The GO analysis of upregulated genes revealed enrichment in biological processes related to nicotinamide nucleotide biosynthesis, pyridine nucleotide biosynthetic process, pyridine-containing compound biosynthetic process, the pyridine nucleotide metabolic process, the nicotinamide nucleotide metabolic process, the oxidoreduction coenzyme metabolic process, and the pyruvate metabolic process (Figure 2C). The GO analysis of downregulated genes revealed enrichment in cell adhesion and biological adhesion processes (Figure 2D).

KEGG analysis of the differentially expressed genes reveals their involvement in various signaling pathways. Upregulated genes were associated with glycolysis/gluconeogenesis, the HIF-1 signaling pathway, carbon metabolism, MicroRNAs in cancer, the biosynthesis of amino acids, the PI3K-Akt signaling pathway, amino sugar and nucleotide sugar metabolism, the biosynthesis of nucleotide sugars, starch and sucrose metabolism, fructose and mannose metabolism, central carbon metabolism in cancer, the glucagon signaling pathway, the pentose phosphate pathway, and the MAPK signaling pathway (Figure 2E). Downregulated genes were associated with Valine, leucine and isoleucine degradation, ECM–receptor interaction, beta-Alanine metabolism, focal adhesion, butanoate metabolism, fatty acid metabolism and nucleotide metabolism (Figure 2F). These findings suggest that hypoxia plays a role in regulating these signaling pathways, which are known to be important for embryonic development.

### 3.3. HIF-1α Is a Key Mediator of the Effects of Hypoxia and CoCl_2_ on MEFs

To further explore the transcription factors (TFs) regulating genes’ responses to hypoxia, we analyzed the TFs of the 1348 genes whose expression levels were upregulated under hypoxia and found that these genes were regulated mainly by p53, SP1, NF-κB, STAT3 and HIF1a (Table 2). HIF1a is a critical TF that responds to hypoxia. We used an HIF1α inhibitor to inhibit HIF1α accumulation, and found that it effectively reduced the accumulation of HIF1α (Figure 3A), decreased the mRNA expression levels of *Pdk1*, *Ldha* and *Glut1* (Figure 3B), and reduced ROS production (Figure 3C,D). Additionally, the HIF1α inhibitor inhibited the invasion and migration of MEFs under hypoxia (Figure 3E–H), but did not inhibit apoptosis (Figure 3I,J). These results indicate that HIF1a plays important roles in the responses of MEFs to hypoxia.

To investigate whether HIF1α mediates the migration and invasion of MEFs induced by low concentrations CoCl_2_, we treated MEFs with low concentrations CoCl_2_, and then measured their migration and invasion after treatment with the HIF1α inhibitor HIF1α-IN-2. We found that the HIF1α inhibitor effectively suppressed the migration and invasion of MEFs induced by CoCl_2_ (Figure 3K–N, Appendix A). Furthermore, we treated MEFs with high concentrations CoCl_2_ to induce ROS production, and then treated them with HIF1α-IN-2. The result show that HIF1α-IN-2 did not decrease ROS production (Figure 3O,P). Finally, we treated MEFs with high concentrations CoCl_2_ to induce apoptosis and then treated them with HIF1α-IN-2. The results show that HIF1α-IN-2 could inhibit CoCl_2_-induced apoptosis (Figure 3Q,R). These findings suggest that HIF1α is a key mediator of the effects of CoCl_2_ on migration, invasion and apoptosis in MEFs.

### 3.4. New Genes in Response to Hypoxia in MEFs

To further explore genes’ responses to hypoxia, we analyzed the genes whose expression levels were upregulated under hypoxia and identified several novel genes, including *Proser2*, *Bean1*, *Dpf1*, *Rnf128*, and *Fam71f1* (Appendix A). These genes have not been traditionally considered to be involved in the hypoxic response. Proser 2 was reported to suppress invasion by increasing the level of p-AMPK in pancreatic ductal adenocarcinoma (PDAC) [32]. Thus, Proser 2 may be involved in the migration of MEFs under hypoxia. Bean1 can interact with NEDD4, which is developmentally regulated, and is highly expressed in embryonic tissues [33,34]. Bean1 is upregulated under hypoxia; this suggests that Bean1 may be associated with embryonic development. Mutations in Bean1 are associated with spinocerebellar ataxia type31 [35]. Rnf128 is an E3 ubiquitin ligase, involved in multiple diseases, such as acute lung injury [36], hepatocellular carcinoma [37], and colorectal cancer (CRC) [38]. Rnf128 also participated in regulating CRC via the PI3K-Akt signaling pathway [38]. The upregulation of Rnf128 under hypoxia in MEFs may participate in regulating embryonic development through the PI3K-Akt signaling pathway. Dpf1 was predicted to be involved in the negative regulation of transcription, nervous system development, and the positive regulation of transcription by RNA polymerase II. The functions of Fam71f1 and Dpf1 have not been studied.

The qPCR results further show that the mRNA expression levels of these genes were increased under hypoxia (Figure 4A). To verify the TFs regulating these genes, we detected their mRNA expression levels under hypoxia in the presence of the HIF1α inhibitor HIF1α-IN-2. The qPCR results show that the expression levels of *Proser2*, *Bean1*, *Dpf1*, *Rnf128*, and *Fam71f1* decreased after HIF1α-IN-2 treatment, indicating that these genes are novel HIF1α-regulated genes (Figure 4B). Similarly, their expression decreased after HIF-1α inhibitor treatment under CoCl_2_ induction (Figure 4C). These findings show that *Proser2*, *Bean1*, *Dpf1*, *Rnf128*, and *Fam71f1* were regulated by HIF1α. 

### 3.5. Transcriptomic Profiles of Normoxic and CoCl_2_ Treatment in MEFs Reveal Common and Divergent Patterns 

To investigate the effects of CoCl_2_ on gene expression and regulation in MEFs, we used mRNA-Seq to analyze the transcriptomes of MEFs under normoxia (*n* = 2) and CoCl_2_ treatment (*n* = 2) groups. The volcano plot revealed that 2660 genes were significantly upregulated in the CoCl_2_ treatment group compared with the normoxia group, while 2991 genes were significantly downregulated (Figure 5A). The heatmap shows significant differences in gene expression profiles between the CoCl_2_ treatment and normoxia groups (Figure 5B). GO analysis of upregulated genes in MEFs under CoCl_2_ induction revealed enrichment in processes related to peptide biosynthesis, the amide biosynthetic process, the peptide metabolic process, the cellular amide metabolic process, the organonitrogen compound biosynthetic process, and translation (Figure 5C). GO analysis of the downregulated genes under CoCl_2_ induction revealed enrichment in processes related to the regulation of transcription, DNA templating, the regulation of RNA metabolic processes, the regulation of nucleic acid-templated transcription, the regulation of RNA biosynthetic processes, the regulation of nucleobase-containing compound metabolic processes, the regulation of biosynthetic processes, and the regulation of macromolecule biosynthetic processes (Figure 5D).

KEGG analysis of the differentially expressed genes reveals their involvement in various signaling pathways. Upregulated genes were mainly associated with oxidative phosphorylation, cholesterol metabolism, and the p53 signaling pathway (Figure 3E). Downregulated genes were mainly associated with ECM–receptor interaction, the PI3K-Akt signaling pathway, the TNF signaling pathway, cytokine–cytokine receptor interaction, the JAK-STAT signaling pathway, and the NOD-like receptor signaling pathway (Figure 3F). Furthermore, the upregulated genes under CoCl_2_ treatment were regulated mainly by Trp53, SP1 and NF-κB (Table 3). These findings indicate that CoCl_2_ can partly mimic hypoxia in MEFs.

### 3.6. Transcriptomic Profiles of Hypoxic and CoCl_2_ Treatment in MEFs Reveal Common and Divergent Patterns 

To reveal the differences in gene regulation between hypoxia and CoCl_2_ treatment, we analyzed the transcriptome of MEFs from hypoxia (*n* = 2) and CoCl_2_ treatment (*n* = 2) groups. The volcano plot reveals that 2272 genes were significantly upregulated in the hypoxia group compared with the CoCl_2_ group, while 1848 genes were significantly downregulated (Figure 6A). The heatmap shows significant differences in gene expression profiles between the hypoxia and CoCl2 treatment groups (Figure 6B). GO analysis of the upregulated genes in the hypoxia group reveals enrichment in processes related to the regulation of transcription, DNA templating, the regulation of RNA metabolic processes, the regulation of nucleic acid-templated transcription, the regulation of RNA biosynthetic processes, the regulation of nucleobase-containing compound metabolic processes, the regulation of biosynthetic processes, and the regulation of macromolecule biosynthetic processes (Figure 6C). GO analysis of the downregulated genes revealed enrichment in processes related to cellular protein catabolic processes, proteolysis involved in cellular protein catabolic processes, protein catabolic processes, cellular macromolecule catabolic processes, macromolecule catabolic processes, organonitrogen compound catabolic processes, peptide biosynthetic processes, and amide biosynthetic processes (Figure 6D).

KEGG analysis of the differentially expressed genes reveals their involvement in various signaling pathways. Upregulated genes were mainly associated with the PI3K-Akt signaling pathway, signaling pathways regulating the pluripotency of stem cells, the insulin signaling pathway, the Rap1 signaling pathway, and the AMPK signaling pathway (Figure 6E). Downregulated genes were mainly associated with oxidative phosphorylation, the p53 signaling pathway, carbon metabolism and multiple disease progresses (Figure 6F).

## 4. Discussion

The development of a single cell into a complex multicellular organism has been studied for centuries [39]. During this intricate process, any deviation from the precisely orchestrated sequence of events can lead to developmental defects, even resulting in embryonic lethality. Neural tube defects (NTDs) in embryonic development lead to brain and/or spinal cord problems [40]. Congenital heart defects (CHDs) can lead to malformation and fetal death [41]. In addition to the diseases mentioned above, other conditions caused by embryonic malformations include cleft lip and palate [42], limb malformations [43], chromosomal abnormalities [44], and single-gene disorders [45]. Therefore, understanding the mechanisms underlying this process remains a crucial area of research.

Embryonic fibroblasts are a diverse population of cells that play a vital role in various developmental processes, including tissue morphogenesis and organogenesis. Embryonic fibroblasts promote cardiomyocyte proliferation through the *β*1 integrin, ERK, and PI3K/Akt pathways, and the cardiomyocyte-specific *β*1 integrin in KO mice leads to embryonic lethality [46]. These results provide a new strategy for heart regenerative therapy. *Ski*^−/−^ mouse embryo fibroblasts exhibit high levels of genome instability [47]. Therefore, understanding the function of embryonic fibroblasts can contribute to elucidating the mechanisms underlying embryonic malformations and identifying potential therapeutic targets.

Hypoxia plays a crucial role in embryonic development, such as organogenesis, cell differentiation, and vascularization [48]. Mammalian embryos develop under hypoxic conditions; a previous study showed that the deletion of HIF1a, a key regulator of the hypoxia response, leads to embryonic lethality [49]. This suggests the critical role of the hypoxia signaling pathway in embryonic development. Although hypoxia is essential for embryonic development, and fibroblasts play a vital role in embryonic development, the specific changes that occur in embryonic fibroblasts under hypoxic remain unclear. Here, we demonstrate that hypoxia can induce multiple changes in MEFs and stimulate multiple signaling pathways.

Animals have evolved a series of processes to respond to hypoxia by activating multiple transcription factors, including the energy and nutrient sensor mTOR [50], the nuclear factor NF-κB transcriptional response [51], and HIF1α [48]. In this paper, in addition to NF-κB and HIF, we identified several other TFs that respond to hypoxia in MEFs, including transformation-related protein 53 (p53), sp1 transcription factor (SP1), signal transducer and activator of transcription 3 (STAT3) and early growth response 1 (Egr1) (Table 2). p53 mutations are reported to cause developmental defects and premature aging phenotypes in humans [52]. Sp1 is required for the maintenance of differentiated cells during early embryonic development [53]. STAT3 maintains the pluripotency of ESCs by regulating the expression of genes associated with pluripotency, such as Oct4, Sox2, and Nanog [54]. STAT3 deficiency disrupts essential cell signaling pathways, leading to developmental abnormalities and embryonic death [55]. Our results indicate that these TFs respond to hypoxia in MEFs, suggesting that they also play important roles in regulating the function of MEFs at the late stage of embryo implantation. The interaction between these TFs and developmental defects will be an interesting area for future study.

HIF1a is one of the most important TFs that respond to hypoxia [56], and HIF1a deficiency leads to embryonic lethality [48]. Our results indicate that HIF1α is activated under hypoxia (Figure 1 and Figure 3A); these results are consisted with those of Randall’s group [57]. Our results indicate that HIF1a plays important roles in MEFs by promoting migration and invasion, and regulating the expression of metabolite genes. The pathways that respond to hypoxia in MEFs are mainly involved in glycolysis/gluconeogenesis and the HIF-1 signaling pathway. In addition, our results suggest that HIF1α is a major TF that responds to hypoxia in MEFs. We also identified several new genes, *Proser2*, *Bean1*, *Dpf1*, *Rnf128*, and *Fam71f1*, that are regulated by HIF1α. Hypoxia triggers the accumulation of HIF1a, which in turn regulates the expression of genes involved in various fibroblast functions. This intricate interplay between hypoxia, HIF1a, and fibroblast activity underscores the importance of oxygen availability and HIF1a signaling in maintaining the diverse roles of MEFs in embryonic development. However, we cannot definitively determine the detailed pathways in which the newly identified genes participate, nor can we ascertain whether their upregulation is beneficial for embryonic development. Future studies will investigate the detailed signaling pathways involved, and the impacts on embryonic development through knockdown and overexpression experiments.

CoCl_2_ is widely employed to simulate hypoxic conditions, providing valuable insights into cellular responses to hypoxia. CoCl_2_ induces hypoxia by stabilizing HIF, inhibiting cellular respiration, inducing oxidative stress, and potentially affecting other pathways, making it a valuable tool for studying cellular responses to hypoxia [58]. However, its effectiveness in mimicking hypoxia may vary depending on cell types. Our results show that CoCl_2_ can promote the invasion and migration of MEFs, induce the production of ROS and apoptosis of MEFs, and promote metabolic reprogramming, which are similar to the effects of hypoxia in MEFs. The differential effects of CoCl_2_ on *Ldha*, *Glut1*, and *Pdk1* are likely due to off-target effects of CoCl_2_. CoCl_2_ is a chemical used to mimic low-oxygen conditions, and it can affect the expression of genes involved in energy metabolism in embryonic fibroblasts. However, it is important to note that CoCl_2_ can also have other effects on cells, which may explain why it affects the expression of some genes differently to others. More research is needed to understand the exact mechanisms behind these differences. These findings indicate that CoCl_2_ can partly mimic hypoxia in MEFs. 

Mechanically, we found that CoCl_2_ increases the expressions of *Glut1*, *Ldha*, and *Pdk1*, which are associated with glucose uptake and metabolism, respectively. While both hypoxia and CoCl_2_ treatment can induce ROS production and apoptosis, the HIF1a inhibitor can suppress hypoxia-induced ROS generation, but not CoCl_2_-induced ROS generation. The HIF1a inhibitor can suppress CoCl_2_-induced apoptosis but not hypoxia-induced apoptosis, suggesting that hypoxia and CoCl_2_ treatment induce apoptosis and ROS through different mechanisms. Previously, a study found that HIF1^−/−^ MEFs prevent apoptosis under hypoxia because of the reduction in ROS via the inhibition of the expression of PDK1 [59]. This research is contrasted with our research, and should be studied in the future. Using mRNA-seq technology, we found that differentially expressed genes (DEGs) in MEFs produced under CoCl_2_ induction are mainly associated with peptide biosynthetic processes, amide biosynthetic processes, peptide metabolic processes, cellular amide metabolic processes, organonitrogen compound biosynthetic processes, and translation. The pathways in MEFs after CoCl_2_ treatment are mainly involved in oxidative phosphorylation, cholesterol metabolism, and the p53 signaling pathway. Furthermore, the upregulated genes produced under CoCl_2_ treatment were mainly regulated by p53 and SP1 (Table 3). These results indicate that CoCl_2_ can only partly mimic hypoxia in MEFs.

In this study, we investigated the effects of hypoxia (both low oxygen and CoCl_2_ treatment) on the phenotype and gene expression of mouse embryonic fibroblasts (MEFs) in vitro. We further explored the involvement of the HIF1α signaling pathway in mediating these effects by inhibiting HIF1α activity. Our findings suggest that HIF1α signaling plays a crucial role in regulating the MEF phenotype and gene expression under hypoxic conditions. Future studies using gene knockdown and overexpression approaches will be conducted to elucidate the detailed molecular mechanisms by which hypoxia regulates the MEF phenotype and gene expression.

## 5. Conclusions

In summary, our data indicate that hypoxia can induce migration and the metabolic reprogramming of MEFs, promote the production of ROS and apoptosis of MEFs, and activate HIF1α, which is a major transcription factor that responds to hypoxia in MEFs. In addition, we identified several new genes, *Proser2*, *Bean1*, *Dpf1*, *Rnf128*, and *Fam71f1*, that are regulated by HIF1α in MEFs. Furthermore, our data suggest that CoCl_2_ can partly mimic hypoxia in MEFs. These findings highlight the importance of HIF1a signaling in maintaining the diverse roles of MEFs in embryonic development.

## Figures and Tables

**Figure 1 biology-13-00598-f001:**
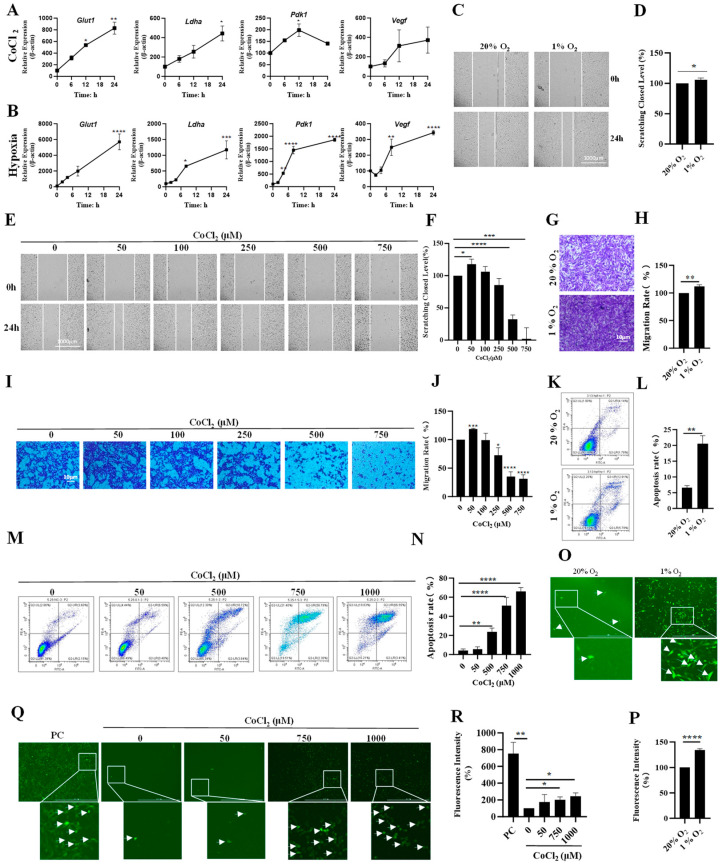
Establishment of hypoxia model with 1% oxygen and CoCl_2_ in MEFs. (**A**) MEFs were treated with 500 μM CoCl_2_ for the indicated time, and the mRNA levels of the indicated genes were analyzed by qPCR (*n* = 3 replicates). (**B**) MEFs were treated with 1% O_2_ for the indicated time, and the mRNA levels of the indicated genes were analyzed by qPCR (*n* = 3 replicates). (**C**,**D**) Images and quantification of migration of MEFs treated with 1% O_2_ for 24 h. Values are expressed as relative to the normoxia control group. Data are presented as the mean ± SEM. *n* = 3 independent experiments. Scale bars: 1000 μm. (**E**,**F**) Migration of MEFs treated with the indicated concentrations of CoCl_2_. Values are expressed as relative to the normoxia control group. Data are presented as the mean ± SEM. *n* = 3 independent experiments. Quantification of wound closure from three independent experiments is shown. Scale bars: 1000 μm. (**G**,**H**) MEFs were subjected to 1% O_2_ for 2 days and then subjected to a cell invasion assay. Quantification of cells in the invaded area from three independent experiments is shown. Scale bars: 10 μm. (**I**,**J**) MEFs were treated with the indicated concentrations of CoCl_2_ for 2 days and then subjected to a cell invasion assay. Quantification of cells in the invaded area from three independent experiments is shown. Scale bars: 10 μm. (**K**,**L**) Images and apoptosis rate of MEFs under hypoxia. Values are expressed as relative to the negative control group. Data are presented as the mean ± SEM. *n* = 3 independent experiments. (**M**,**N**) Images and apoptosis rate of MEFs in the presence of CoCl_2_. Values are expressed as relative to the negative control group. Data are presented as the mean ± SEM. *n* = 3 independent experiments. (**O**,**P**) MEFs were subjected to hypoxia for 24 h and then subjected to a ROS production assay. Quantification of green fluorescence intensity from three independent experiments is shown. Scale bars: 1000 μm. (**Q**,**R**) MEFs were treated with the indicated concentrations of CoCl_2_ for 24 h and then subjected to a ROS production assay. Quantification of green fluorescence intensity from three independent experiments is shown. PC: positive control. Scale bars: 1000 μm. The white arrows in (**O**,**Q**) represent cells that are producing ROS cells. * *p* < 0.05, ** *p* < 0.01, *** *p* < 0.0001, **** *p* < 0.0001.

**Figure 2 biology-13-00598-f002:**
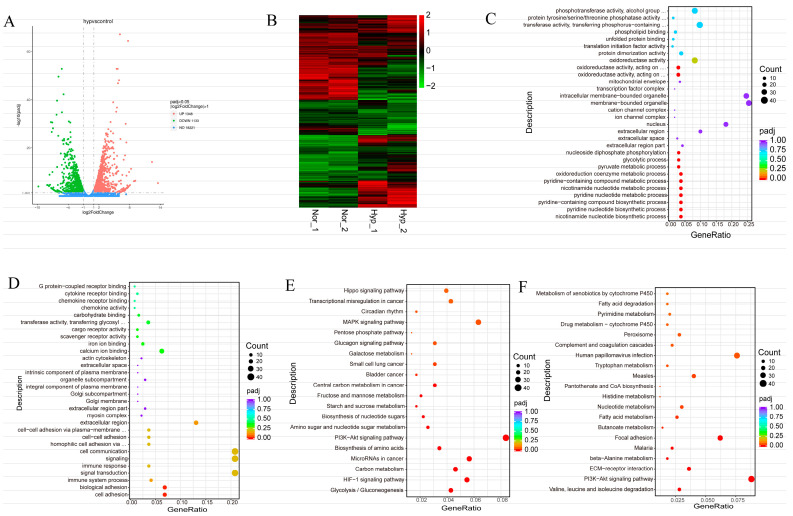
The functional annotation of the differentially expressed genes in MEFs’ response to hypoxia. (**A**,**B**) Volcano map (**A**) and heatmap (**B**) of the differentially expressed genes under hypoxia. (**C**–**F**) The GO annotation (upregulation in (**C**) and downregulation in (**D**)) and KEGG pathway enrichment (upregulation in (**E**) and downregulation in (**F**)) analysis of the differentially expressed genes. Nor: normoxia group. Hyp: hypoxia was induced in 1% O_2_.

**Figure 3 biology-13-00598-f003:**
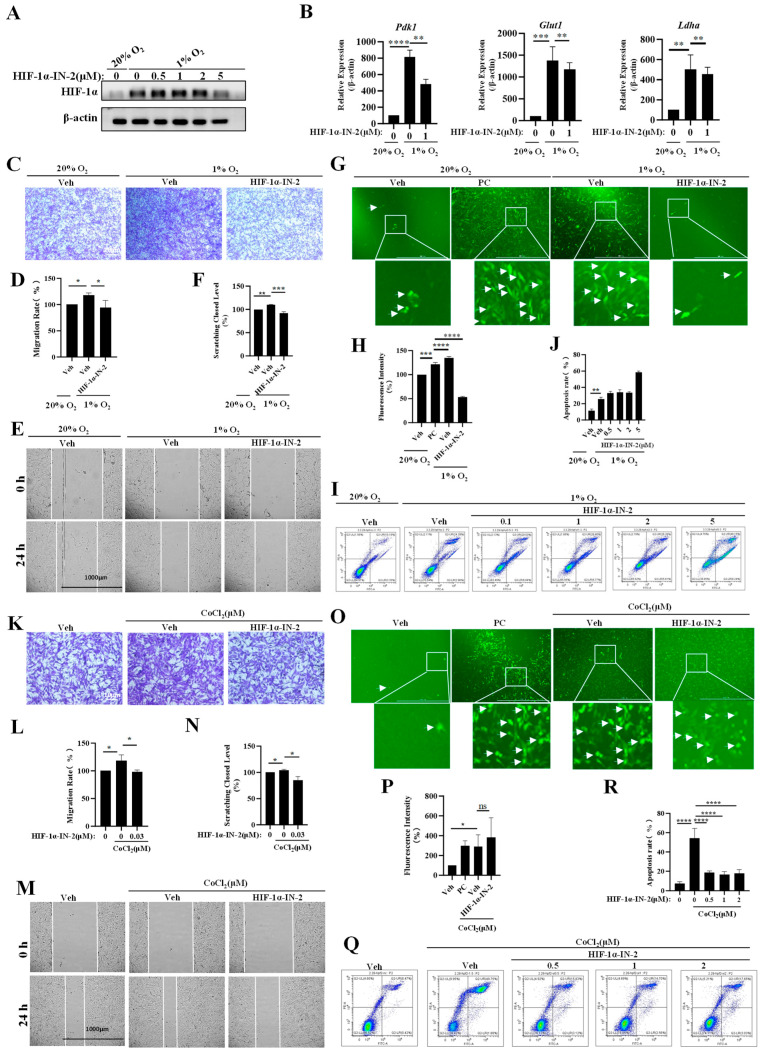
A HIF1α inhibitor inhibits multiple processes in MEFs. (**A**) MEFs were treated with 1% O_2_ in the presence or absence of HIF-1**α**-IN2 for the indicated time, and the protein levels of the indicated proteins were analyzed by Western blotting (*n* = 3 replicates) (Appendix A: uncropped gels). (**B**) MEFs were treated with 1% O_2_ in the presence or absence of HIF-1**α**-IN2 for the indicated time, and the mRNA levels of indicated genes were analyzed by qPCR (*n* = 3 replicates). (**C**,**D**) MEFs were treated with 1% O_2_ in the presence or absence of HIF-1**α**-IN2 for the indicated time and then subjected to cell invasion assay. Quantification of cells in the invaded area from three independent experiments is shown. Scale bars: 10 μm. (**E**,**F**) Images and quantification of migration of MEFs under 1% O_2_ in the presence or absence of HIF-1**α**-IN2 for 24 h. Values are expressed as relative to the negative control group. *n* = 3 independent experiments. Scale bars: 1000 μm. (**G**,**H**) MEFs were subjected to 1% O_2_ for 24 h in the presence or absence of HIF-1α-IN2 and then subjected to a ROS production assay. Quantification of green fluorescence intensity from three independent experiments is shown. PC: positive control. Scale bars: 1000 μm. (**I**,**J**) Images and apoptosis rates of MEFs under 1% O_2_ in the presence or absence of HIF-1α-IN2. Values are expressed as relative to the negative control group. Data are presented as the mean ± SEM. *n* = 3 independent experiments. (**K**,**L**). MEFs were treated with 50 μM CoCl_2_ in the presence or absence of HIF-1α-IN2 for the indicated time, and then subjected to cell invasion assay. Quantification of cells in the invaded area from three independent experiments is shown. Scale bars: 25 μm. (**M**,**N**) Images and quantification of migration of MEFs under 50 μM CoCl_2_ stimulation in the presence or absence of HIF-1α-IN2 for 24 h. Values are expressed as relative to the negative control group. Data are presented as the mean ± SEM. *n* = 3 independent experiments. Scale bars: 1000 μm. (**O**,**P**) MEFs were subjected to 500 μM CoCl_2_ stimulation for 24 h in the presence or absence of HIF-1α-IN2 and subjected to a ROS production assay. Quantification of green fluorescence intensity from three independent experiments is shown. PC: positive control. Scale bars: 1000 μm. (**Q**,**R**) Images and apoptosis rate of MEFs under 500 μM CoCl_2_ stimulation in the presence or absence of HIF-1α-IN2. Values are expressed as relative to the negative control group. Data are presented as the mean ± SEM. *n* = 3 independent experiments. The white arrows in (**G**,**O**) represent cells that are producing ROS cells. * *p* < 0.05, ** *p* < 0.01, *** *p* < 0.0001, **** *p* < 0.0001.

**Figure 4 biology-13-00598-f004:**
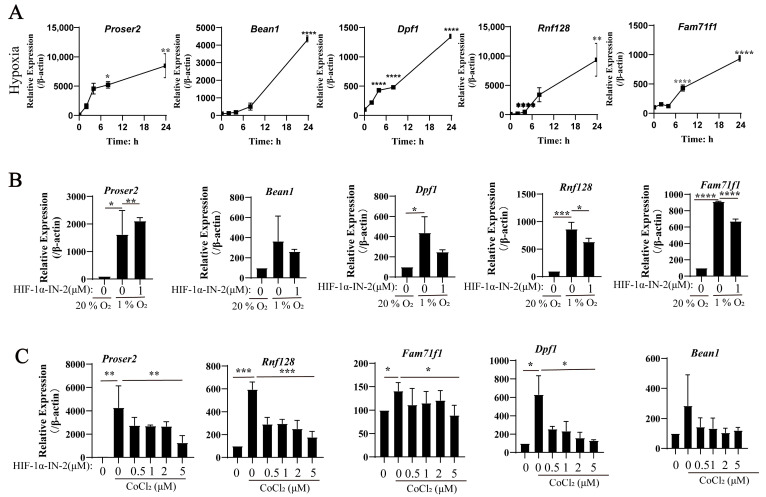
New genes response to hypoxia. (**A**) MEFs were treated with 1% O_2_ for the indicated time, and the mRNA levels of indicated genes were analyzed by qPCR. (*n* = 3 replicates). (**B**) MEFs were treated with 1% O_2_ in the presence or absence of HIF-1**α**-IN2 for the indicated time, and the mRNA levels of the indicated genes were analyzed by qPCR (*n* = 3 replicates). (**C**) MEFs were treated with 500 μM CoCl_2_ in the presence or absence of HIF-1α-IN2 for the indicated time, and the mRNA levels of the indicated genes were analyzed by qPCR (*n* = 3 replicates). * *p* < 0.05, ** *p* < 0.01, *** *p* < 0.0001, **** *p* < 0.0001.

**Figure 5 biology-13-00598-f005:**
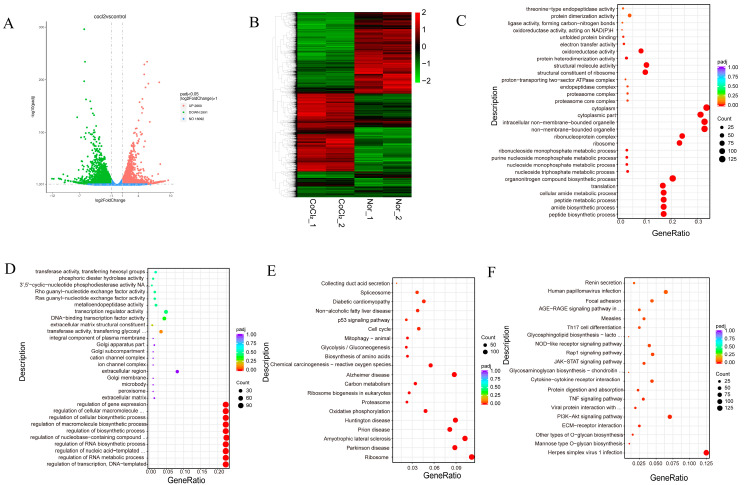
The functional annotation of the differentially expressed genes in MEFs in response to CoCl_2_. (**A**,**B**) Volcano map (**A**) and heatmap (**B**) of the differentially expressed genes after CoCl_2_ treatment. (**C**–**F**) The GO annotation (upregulation in (**C**) and downregulation in (**D**)) and KEGG pathway enrichment (upregulation in (**E**) and downregulation in (**F**)) analysis of the differentially expressed genes. Nor: normoxia group.

**Figure 6 biology-13-00598-f006:**
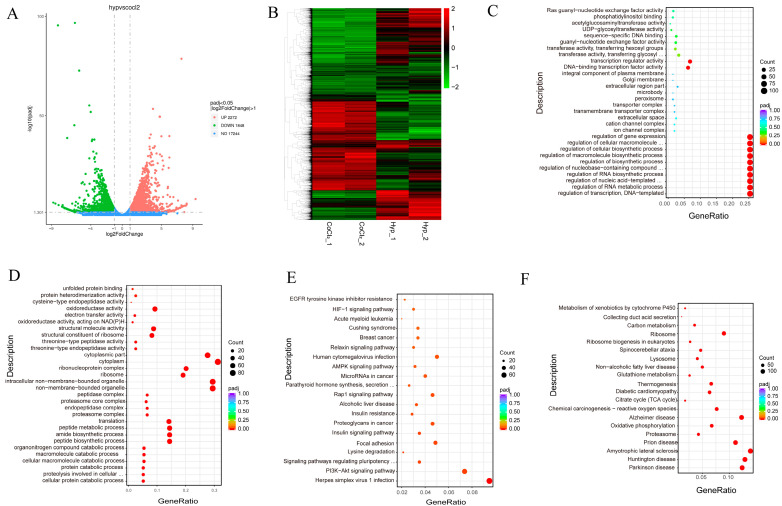
The functional annotation of the differentially expressed genes in MEFs in response to hypoxia and CoCl_2_. (**A**,**B**) Volcano plot (**A**) and heatmap (**B**) of the differentially expressed genes between hypoxia and CoCl_2_ treatment. The volcano plot displays the log 2-fold change versus the -log10 *p*-value for each gene. The heatmap shows the expression levels of differentially expressed genes across different samples. (**C**–**F**) GO annotation (upregulation in (**C**) and downregulation in (**D**)) and KEGG pathway enrichment (upregulation in (**E**) and downregulation in (**F**)) analysis of differentially expressed genes. Hyp: hypoxia group.

**Table 1 biology-13-00598-t001:** Primers in this paper.

Gene	Sequence (5′-3′)
*Bean1-F*	GCACGATACAACCGTACCAG
*Bean2-R*	ACCACACCTATGACAGCGTTC
*Proser2-F*	GAGGTGGCAGTCTGGAGAG
*Proser3-R*	AGAACAGGATAACGTCCTTTTCC
*Pdk1-F*	GGACTTCGGGTCAGTGAATGC
*Pdk1-R*	TCCTGAGAAGATTGTCGGGGA
*Ldha-F*	CAAAGACTACTGTGTAACTGCGA
*Ldha-R*	TGGACTGTACTTGACAATGTTGG
*Dpf1-F*	TTGCTGGAGTTTCCGCATGAT
*Dpf2-R*	TACGGCTTGTCTCGGTCCT
*Rnf128-F*	ATTCAAAGAGGCATCCAAGTCAC
*Rnf129-R*	TGCATTTCGTAATCTTCGAGCAG
*Fam71f1-F*	ACCTTCCCTTCCTTGAATGCC
*Fam71f2-R*	GCAGAGTAGTCCTTCCTCCAC
*Glut1-F*	GCAGTTCGGCTATAACACTGG
*Glut1-R*	GCGGTGGTTCCATGTTTGATTG
*Vegf-F*	CTGCCGTCCGATTGAGACC
*Vegf-R*	CCCCTCCTTGTACCACTGTC
*Actin-F*	GGCTGTATTCCCCTCCATCG
*Actin-R*	CCAGTTGGTAACAATGCCATGT

**Table 2 biology-13-00598-t002:** TFs that upregulate genes in response to hypoxia.

Key TF	# of Overlapped Genes	*p* Value	Q Value	List of Overlapped Genes
p53	44	1.97 × 10^−48^	3.11 × 10^−46^	*Polk*, *E2f1*, *Gdf15*, *Ptgs2*, *Nupr1*, *Klf4*, *Fos*, *Mdm2*, *Rps6ka1*, *Lif*, *Elf4*, *Rrm2b*, *Pcna*, *Btg2*, *Foxo3*, *Slc19a2*, *Mcm7*, *Cdkn1a*, *Rela*, *Atf3*, *Hmox1*, *Dusp4*, *Pmaip1*, *Krt19*, *Klf2*, *Mt1*, *Ankrd1*, *Myc*, *Gabre*, *Srebf1*, *E4f1*, *Dusp1*, *Casp6*, *Plk3*, *Bbc3*, *Adgrb1*, *Cdkn1b*, *Ddit4*, *Serpine1*, *Egr1*, *Igf1r*, *Ndrg1*, *Mxi1*, *Id2*
Sp1	38	8.07 × 10^−31^	6.38 × 10^−29^	*Smad7*, *Myc*, *Dck*, *Pld2*, *Fos*, *Mgarp*, *Bhlhe40*, *Serpine1*, *Acsbg1*, *Srebf1*, *Slc7a5*, *Plaur*, *Egr1*, *Ccnd3*, *Timp3*, *Slc2a1*, *Rnf141*, *Tec*, *Klf16*, *Epor*, *Tnfaip3*, *Cdkn2d*, *Ndrg1*, *Slc2a3*, *Hinfp*, *Ptgs2*, *Ets1*, *Rest*, *Ccnd2*, *Cdkn1a*, *Jun*, *Ecm1*, *Jund*, *Cdh5*, *Igfbp3*, *Mrc1*, *Bsg*, *Itga6*
Nfkb1	34	1.81 × 10^−30^	9.53 × 10^−29^	*Ntn1*, *Nfkbia*, *Pmaip1*, *Ptgs2*, *Nos2*, *Tgfb1*, *Myoz2*, *Ppargc1a*, *Cdkn1a*, *Adora2b*, *Ccnd3*, *Stk39*, *Slc2a1*, *Gadd45b*, *Lif*, *Myc*, *Klf5*, *Fos*, *Per1*, *Tnfaip3*, *Msx1*, *Dusp1*, *Wt1*, *Timp3*, *Sox9*, *Banp*, *Plin2*, *Kdm2a*, *Yy1*, *Plaur*, *Junb*, *Hmox1*, *Ppp1r13l*, *Tec*
Stat3	16	3.85 × 10^−16^	1.52 × 10^−14^	*Klf4*, *Epor*, *Tgfb1*, *Cdkn2d*, *Nos2*, *Ccnd3*, *Mt1*, *Ccnd2*, *Cdkn1b*, *Eed*, *Srebf1*, *Egr1*, *Rela*, *Fos*, *Myc*, *Lif*
Hif1a	10	7.71 × 10^−15^	2.44 × 10^−13^	*Fam162a*, *Mt1*, *Hey1*, *Hey2*, *Hmox1*, *Serpine1*, *Ptgs2*, *Bsg*, *Pfkfb3*, *Ctgf*
Rela	17	2.73 × 10^−14^	7.19 × 10^−13^	*Tnfaip3*, *Myc*, *Klf5*, *Stk39*, *Fos*, *Tec*, *Pecam1*, *Ptgs2*, *Wt1*, *Cxcr4*, *Gabre*, *Cdkn1b*, *Pgf*, *Yy1*, *Nos2*, *Sox9*, *Kdm2a*
Egr1	14	4.19 × 10^−14^	9.46 × 10^−13^	*Cacna1h*, *Cdkn2d*, *Smad7*, *Itga7*, *Rcan1*, *Serpine1*, *Jun*, *Bnip3*, *Ndrg1*, *Ccnd2*, *Ltb*, *Jund*, *Igf1r*, *Gadd45b*
Myc	12	1.77 × 10^−13^	3.49 × 10^−12^	*Myc*, *Rhoa*, *Ndrg1*, *Hnrnpa1*, *Ccnd2*, *Suz12*, *Mxi1*, *Zfp36*, *Timp3*, *Cdkn1a*, *Bcat1*, *Eed*
Cebpb	11	8.2 × 10^−13^	1.44 × 10^−11^	*Cdkn1a*, *Ptgs2*, *Myc*, *Atf3*, *Btg2*, *Cxcr4*, *Cyr61*, *Id2*, *Serpine1*, *Ppargc1a*, *Fos*
Jun	16	1.48 × 10^−12^	2.34 × 10^−11^	*Plin2*, *Plaur*, *Ccnd2*, *Jun*, *Ccnd3*, *Ptgs2*, *Fos*, *Eno2*, *Slc2a1*, *Pdk1*, *Nos2*, *Serpine1*, *Tgfb1*, *Fosl1*, *Hmox1*, *Ppp1r3b*
Trp73	8	1.44 × 10^−11^	2.07 × 10^−10^	*Hey2*, *Pmaip1*, *Gls2*, *Mdm2*, *Cdkn1a*, *Bbc3*, *E2f1*, *Foxo3*
Ep300	11	5.5 × 10^−11^	7.25 × 10^−10^	*Klf5*, *Ldha*, *Txnip*, *Fos*, *Ptgs2*, *Cryab*, *Mt1*, *Hmox1*, *Nos2*, *Cdkn1a*, *Hinfp*
Clock	7	1.17 × 10^−10^	1.42 × 10^−9^	*Cry1*, *Serpine1*, *Nampt*, *Bhlhe41*, *Bhlhe40*, *Per1*, *Cry2*
Crebbp	9	5.84 × 10^−10^	6.6 × 10^−9^	*Fos*, *Nfyb*, *Cryab*, *Fosb*, *Hmox1*, *Hinfp*, *Mt1*, *Per1*, *Ldha*
Creb1	8	1.34 × 10^−9^	1.41 × 10^−8^	*Fos*, *Noct*, *Ppargc1a*, *Ptgs2*, *Slc2a3*, *Ldha*, *Hspa5*, *Nmnat2*
Foxo1	9	3.17 × 10^−9^	3.13 × 10^−8^	*Ccng2*, *Cdkn1a*, *Cdkn1b*, *Ppargc1a*, *Trib3*, *Egr1*, *Srebf1*, *Ccnd2*, *Klf2*
Nfe2l2	9	6.58 × 10^−9^	6.12 × 10^−8^	*Mcm7*, *Txnip*, *Areg*, *Fos*, *Sqstm1*, *Mdm2*, *Hmox1*, *Mt1*, *Tgfb1*
Dmtf1	5	9.01 × 10^−9^	7.91 × 10^−8^	*Junb*, *Ccnd2*, *Ets1*, *Egr1*, *Areg*
Fos	9	1.09 × 10^−8^	9.09 × 10^−8^	*Jun*, *Tgfb1*, *Bdnf*, *Ppp1r3b*, *Tinagl1*, *Fos*, *Mt1*, *Egr1*, *Nos2*
Rb1	6	6.13 × 10^−8^	0.000000484	*Fosl1*, *Cdkn1a*, *Mybl2*, *Fos*, *Jun*, *E2f1*
Myod1	8	7.59 × 10^−8^	0.000000571	*Smad7*, *Fosl1*, *Cdkn1a*, *Itga7*, *Ppargc1a*, *Rb1*, *Ccnd3*, *Myh7b*

p53: Transformation-related protein 53. SP1: Sp1 transcription factor. *NFKB1*: nuclear factor of kappa light polypeptide gene enhancer in B-cells 1. STAT3: signal transducer and activator of transcription 3 (acute-phase response factor). HIF1A: hypoxia inducible factor 1, alpha subunit (basic helix-loop-helix transcription factor). RELA: v-rel reticuloendotheliosis viral oncogene homolog A (avian). Egr1: early growth response 1. MYC: v-myc myelocytomatosis viral oncogene homolog (avian). Cebpb: CCAAT/enhancer binding protein (C/EBP), beta. JUN: jun proto-oncogene. Trp73: Transformation-related protein 73. EP300: E1A binding protein p300. Clock: circadian locomotor output cycles kaput. Crebbp: CREB binding protein. CREB1: cAMP responsive element binding protein 1. Foxo1: fork head box O1. Nfe2l2: nuclear factor, erythroid derived 2, like 2. Dmtf1: cyclin D binding myb-like transcription factor 1. Fos: osteosarcoma oncogene. Rb1: retinoblastoma 1. Myod1: myogenic differentiation 1. #: The numbers of genes regulated by indicated transcription factor.

**Table 3 biology-13-00598-t003:** TFs that upregulate genes in response to CoCl_2_.

Key TF	# of Overlapped Genes	*p* Value	Q Value	List of Overlapped Genes
Trp53	18	0.000000132	0.00000818	*E2f1*, *Gdf15*, *Nupr1*, *Chek1*, *Klf4*, *Mdm2*, *Gtse1*, *Pcna*, *Bax*, *Slc19a2*, *Mcm7*, *Hspa1b*, *Afp*, *Mt1*, *Serpine1*, *Ndrg1*, *Id2*, *Birc5*
Sp1	21	0.00000288	0.0000893	*Serpine1*, *Sirt1*, *Slc7a5*, *Slc2a1*, *Slc3a2*, *Tec*, *Prdx6*, *Klf16*, *Apoe*, *Mertk*, *Cebpa*, *Cdk5r1*, *Ndrg1*, *Hspa1b*, *Nrgn*, *Jun*, *Tspo*, *Tgm1*, *Pemt*, *Bsg*, *Itga6*
Egr1	9	0.0000523	0.00108	*Wnt4*, *Serpine1*, *Cdk5r1*, *Bax*, *Jun*, *Birc5*, *Nr4a1*, *Hsd11b2*, *Ndrg1*
Nfe2l2	7	0.000276	0.00427	*Mcm7*, *Gclc*, *Slc7a11*, *Abcc4*, *Srxn1*, *Mdm2*, *Mt1*
Nfkb1	14	0.000385	0.00478	*Ntn1*, *Mmp3*, *Birc5*, *Phex*, *Cebpa*, *Apoe*, *Gclm*, *Slc2a1*, *Klf5*, *Per1*, *Amh*, *Plin2*, *Hsd11b2*, *Tec*
Trp73	4	0.000988	0.01	*Gls2*, *Mdm2*, *E2f1*, *Bax*
Myc	6	0.00113	0.01	*Bax*, *Wnt4*, *Ndrg1*, *Nop56*, *Zfp36*, *Nrgn*
Jun	10	0.00134	0.0103	*Plin2*, *Gclc*, *Jun*, *Tgm1*, *Mmp3*, *Slc2a1*, *Serpine1*, *Phex*, *Sirt1*, *Srxn1*
Rbl2	3	0.00149	0.0103	*Birc5*, *Stmn1*, *Pcna*
Nr1i2	3	0.00264	0.0164	*Insig1*, *Cyp24a1*, *E2f1*
Sp3	7	0.00399	0.0216	*Nrgn*, *Bsg*, *Itga6*, *Tspo*, *Mertk*, *Cdk5r1*, *Cebpa*

Trp53: Transformation-related protein 53. SP1: Sp1 transcription factor. Egr1: early growth response 1. Nfe2l2: nuclear factor, erythroid derived 2, like 2. NFKB1: nuclear factor of kappa light polypeptide gene enhancer in B-cells 1. Trp73: Transformation-related protein 73. MYC: v-myc myelocytomatosis viral oncogene homolog (avian). JUN: jun proto-oncogene. Rbl2: retinoblastoma-like 2. Nr1i2: nuclear receptor subfamily 1, group I, member 2. Sp3: rans-acting transcription factor 3. #: The numbers of genes regulated by indicated transcription factor.

## Data Availability

All data generated or analyzed during this study are included in this published article (and its Appendix A).

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
