# Peer review of "Hypoxia-Induced Adaptations of Embryonic Fibroblasts: Implications for Developmental Processes"

_biology, 2024, doi:10.3390/biology13080598_

Round 1

Reviewer 1 Report

Comments and Suggestions for Authors

The manuscript by Li et al. presents a comprehensive study on how hypoxia induces adaptations of embryonic fibroblasts. The authors are commended for an interesting study. Specific comments/suggestions to further improve the manuscript prior to acceptance are as follows:

·       Methods

1.     Please add the methods for sequencing performed in this research.

2.    Please add a detailed description of the image analysis process used for the detection of ROS.

3. In terms of statistical analysis, it has been observed that several experiments involve more than two groups. In such cases, it would be more appropriate to utilize ANOVA for the analysis.

·       Results

1.     In general, the authors should enhance their writing to provide a more comprehensive and clear description of the results. And some of the figure captions should be revised.

2.     Figure 1.

o   Figure 1E and F appear to be a discrepancy in the concentrations of CoCl2 used. Figure 1E indicates that the concentrations used were 0, 0.1, 0.2, 0.5, 1, and 2 µM CoCl2. However, the plots in Figure 1F suggest that the concentrations used were 0, 0.1, 0.2, 0.5, 1, and 1.5 µM CoCl2. Please clarify this inconsistency.

o   Figure 1I and J, the authors included 0.05 µM CoCl2 and excluded the 0.2 µM  group, and showed that 0.05 µM CoCl2 significantly increased cell migration, please clarify why this new concentration was included and implicate this result. What is the effect of 0.05 µM CoCl2 on cell viability?

o   Figure 1L and R, please check the labeling of the x-axis.

3.     Section 3.2, the authors should provide more details about the conditions for the hypoxia group. Was the hypoxic state achieved through low oxygen levels or was it induced using CoCl2?

4.     Figure 3.

o   Please check the labeling of some panels, what is PC?

o   Why did the authors use 0.03 uM HIF-1α-IN2 in Figure 3L and M? If the concern is that a higher dose might induce cell apoptosis, they should include data showing how this specific concentration impacts cell viability.

5.     Figure 4. Please clarify the concentration of CoCl2 that was used and revise the caption.

6.     What are the effects of HIF-1α-IN2 on healthy cells?

·       Discussion

1.     Could the authors discuss any potential limitations and future directions that have been identified in this study?

Comments on the Quality of English Language

The quality of the English writing should be enhanced.

Author Response

Overall Response: We would like to express our sincere gratitude to all reviewers for their valuable time and insightful comments on our manuscript. We deeply appreciate the constructive feedback you have provided, which has been instrumental in improving the quality of our work. In this point-by-point response letter, reviewer's comments were marked in dark blue italics, followed by our detailed response. All revisions and supplementary new text are clearly highlighted in red.

Reviewer #1:

The manuscript by Li et al. presents a comprehensive study on how hypoxia induces adaptations of embryonic fibroblasts. The authors are commended for an interesting study. Specific comments/suggestions to further improve the manuscript prior to acceptance are as follows:

 Methods

1.Please add the methods for sequencing performed in this research.

Response:

Thank you for your question. We have added a detailed description of the sequencing methods used in the study to the revised materials and methods section of the revised manuscript on page 4, line 164 to line 191.

page 4, line 164 to line 191.

The RNA extraction and RNA sequencing (RNA-seq) methods used in this study are identical to those described in a previous publication [1]. Briefly, RNA integrity was assessed using the RNA Nano 6000 Assay Kit on the Bioanalyzer 2100 system (Agilent Technologies, CA, USA) [2]. mRNA was subsequently purified, followed by first-strand cDNA synthesis. Second-strand cDNA synthesis was then performed using DNA Polymerase I and RNase H [3]. After adenylation of the 3' ends of DNA fragments, adaptors with hairpin loop structures were ligated to prepare for hybridization. The library fragments were purified using the AMPure XP system [4]. PCR was then performed with Phusion High-Fidelity DNA polymerase, universal PCR primers, and index (X) primers. Finally, PCR products were purified (AMPure XP system) and library quality was assessed on the Agilent Bioanalyzer 2100 system. The clustering of the index-coded samples was performed on a cBot Cluster Generation System using the TruSeq PE Cluster Kit v3-cBot-HS (Illumina) according to the manufacturer's instructions [5]. After cluster generation, the library preparations were sequenced on an Illumina NovaSeq platform, generating 150 bp paired-end reads.

Raw data (fastq format) was processed using the fastp software. Clean data (clean reads) were obtained by removing reads containing adapters, reads containing poly-N sequences, and low-quality reads. The reference genome index was built using Hisat2 v2.0.5, and paired-end clean reads were aligned to the reference genome using Hisat2 v2.0.5 [6]. The mapped reads of each sample were assembled by StringTie (v1.3.3b) using a reference-based approach [7]. Feature Counts v1.5.0-p3 was used to count the reads mapped to each gene [8]. FPKM for each gene was then calculated based on the gene length and the number of reads mapped to that gene. Differential expression analysis[9], GO and KEGG enrichment analysis of differentially expressed genes were performed using the cluster Profiler R package, with gene length bias correction [10, 11]. GO terms with a corrected P-value less than 0.05 were considered significantly enriched by differentially expressed genes. The raw data from our RNA-sequencing experiments have been deposited in the GEO database (accession number: PRJNA1134336)

  1. Please add a detailed description of the image analysis process used for the detection of ROS.

Response:

Thank you for your question. We have added a detailed description of the image analysis process used for the detection of ROS to the revised materials and methods section of the revised manuscript on page 3, line 107 to line 113.

page 3, line 107 to line 113.

2',7'- dichlorofluorescein (DCF) was measured under Cytation1 (Bio-Tek; Winooski, VT, USA) with spectra of 469 nm excitation/525 nm emission. The fluorescence intensity was measured by Image J. Briefly, images were opened in ImageJ software. Background subtraction was performed using the rolling ball algorithm to remove non-specific signals. The intensity of the ROS signal was measured. All ROS intensity values were normalized to the intensity of the control group to account for differences in dye concentration.

  1. 3. In terms of statistical analysis, it has been observed that several experiments involve more than two groups. In such cases, it would be more appropriate to utilize ANOVA for the analysis.

Response:

We appreciate the reviewer's feedback and acknowledge the importance of providing clear and consistent details about our statistical analysis methods. We have re-analyzed our experimental data and have added detailed statistical methods to the revised materials and methods section on page 5, line 195 to line 204.

page 5, line 194 to line 203.

Data normality and homogeneity of variance were assessed using the Shapiro-Wilk. Two-group comparisons employed t-tests for equal variances and normal distributions, Wilcoxon tests for equal variances and non-normal distributions, and Welch's t-tests for unequal variances and normal distributions. For more than two independent groups, one-way ANOVAs with Tukey's post hoc tests were used for equal variances and normal distributions, while Kruskal-Wallis tests with Dunn's post hoc tests were employed for equal variances and non-normal distributions. Welch's ANOVAs with Games-Howell post hoc tests were applied for unequal variances and normal distributions. A significance level of P < 0.05 was adopted for all analyses. All statistical procedures were conducted using Prism 10.

Results

1.In general, the authors should enhance their writing to provide a more comprehensive and clear description of the results. And some of the figure captions should be revised.

Response:

Thank you for your feedback. We have thoroughly reviewed the manuscript and made significant efforts to improve the clarity and comprehensiveness of the results section. We have also revised the figure captions to ensure they accurately reflect the data presented.

  1. 2. Figure 1.

1)  Figure 1E and F appear to be a discrepancy in the concentrations of CoCl2 used. Figure 1E indicates that the concentrations used were 0, 0.1, 0.2, 0.5, 1, and 2 µM CoCl2. However, the plots in Figure 1F suggest that the concentrations used were 0, 0.1, 0.2, 0.5, 1, and 1.5 µM CoCl2. Please clarify this inconsistency.

Response:

Thank you for bringing this discrepancy to our attention. We apologize for the confusion caused by the inconsistent labeling of the CoCl2 concentrations in Figure 1E and F. We have carefully reviewed our data and confirmed that the correct concentrations used in the experiment were 0, 50, 100, 250, 500, and 750 µM CoCl2. The original figure legend in Figure 1E incorrectly listed the highest concentration as 2 µM CoCl2. We have corrected this error in the revised Figure 1E and 1F, which now accurately reflects the actual concentrations used.

2)  Figure 1I and J, the authors included 0.05 µM CoCl2 and excluded the 0.2 µM group, and showed that 0.05 µM CoCl2 significantly increased cell migration, please clarify why this new concentration was included and implicate this result. What is the effect of 0.05 µM CoCl2 on cell viability?

3) Figure 1L and R, please check the labeling of the x-axis.

Response:

Thank you for bringing this discrepancy to our attention. We apologize for the inconsistencies in Figure 1I, 1J, 1L and 1R. We have carefully reviewed our data and confirmed the following corrections:

Figure 1I: The concentration of CoCl2 used in this experiment was 50 and 100 µM, not 0.05 µM as incorrectly labeled in the Figure1I and J. We have corrected the label to accurately reflect the actual concentration used in revised Figure 1I and J.

We have conducted additional experiments to assess the effect of CoCl2 on cell viability. Our results indicate that 50 µM does not significantly affect cell viability in MEFs. We have now compiled the data into revised Figure S1A and described them in revised text on page 5, line 212.

We have attached the revised Figures 1L and R with the corrected labels in revised Figure 1L and R.

page 5, line 212.

and inhibit the proliferation of MEFs (Figure S1A, S1B).

  1. 3.     Section 3.2, the authors should provide more details about the conditions for the hypoxia group. Was the hypoxic state achieved through low oxygen levels or was it induced using CoCl2?

Response:

Thank you for your question. We apologize for the lack of clarity in our manuscript regarding the method used to induce hypoxia. The hypoxic state was achieved by exposing the cells to a controlled environment with a reduced oxygen concentration of 1% for 24 h, which directly mimics the physiological conditions of uterine environment. Control cells were cultured in a normoxia environment (21% O2) for the same period. We have now added a detailed description of the hypoxia induction method in revised Section 3.2 on page 7, line 243 to line 243.

page 7, line 243 to line 244.

the hypoxic state was induced by exposing MEFs to a controlled environment with a reduced oxygen concentration of 1% for 24 h.

  1. 4.     Figure 3.

1)  Please check the labeling of some panels, what is PC?

2) Why did the authors use 0.03 uM HIF-1α-IN2 in Figure 3L and M? If the concern is that a higher dose might induce cell apoptosis, they should include data showing how this specific concentration impacts cell viability.

Response:

Thank you for bringing this to our attention. We apologize for the lack of clarity in the labeling of "PC" in Figure 3. We have carefully reviewed the Figure 3 and confirmed that "PC" stands for "positive control". We have revised Figure 3 to include a more explicit explanation of the "PC" label in the Revised Figure legends on page 11, line 308 and line 319.

We chose to use 0.03 µM HIF-1α-IN2 in Figure 3L and M because we observed a significant increase in cell migration at this concentration without any apparent cell death. We wanted to focus on the effects of HIF-1α-IN2 on cell migration under conditions where cell viability was not compromised. During our cell migration experiments, we indeed observed that high concentrations of HIF-1α-IN2 induced apoptosis in a portion of the cells. However, when using concentrations of 0.03, 0.05 and 0.1 µM, we did not observe any significant cell death. While we have only presented the results for 0.03 µM in the main manuscript, we have included the migration results for both low and high concentrations in Revised Figure S1D.

We have conducted additional experiments to assess the effect of HIF-1α-IN2 on cell viability at higher concentrations (0, 0.01, 0.03, 0.05, 0.1, 0.5, 1, and 2 µM). Our results indicate that 0.01, 0.03, and 0.05 µM does not significantly affect cell viability in MEFs. We have included this data in the Revised Figure S1C.

page 11, line 308 and line 319.

PC: positive control;

  1. 5.     Figure 4. Please clarify the concentration of CoCl2 that was used and revise the caption.

Response:

Thank you for pointing out this omission. We apologize for the lack of clarity in the caption of Figure 4. We have carefully reviewed the figure and confirmed that the concentration of CoCl2 used in the experiment was 500 µM. We have revised the caption in revised manuscript on page 12, line 367.

page 12, line 367.

(C), MEFs were treated with 500 μM CoCl2 in the presence or absence of HIF-1α-IN2 for the indicated time, and the mRNA levels of the indicated genes were analyzed by qPCR (n = 3 replicates).

  1. What are the effects of HIF-1α-IN2 on healthy cells?

Response:

Thank you for your question. We have conducted additional experiments to investigate the effects of HIF-1α-IN2 on healthy cells. We used a CCK-8 assay to assess the viability of healthy cells treated with different concentrations of HIF-1α-IN2, and found that HIF-1α-IN2 did not significantly affect the viability of healthy MEFs at 0.01-0. 05uM (Revised Figure S1C). We used a scratch wound assay to assess the migration of healthy cells treated with different concentrations of HIF-1α-IN2, and found that HIF-1α-IN2 did not significantly affect the migration of healthy cells at low concentrations (0.01-0.05uM) (revised Figure S2).

 Discussion

  1. 1.     Could the authors discuss any potential limitations and future directions that have been identified in this study?

Response:

We appreciate the reviewer's suggestion and acknowledge the importance of discussing the limitations of our study and potential future research directions. We have addressed this point in the revised manuscript by adding potential limitations and future directions to the discussion specifically focusing on these aspects on page 15, line 479 to line 480; page 16, page 16, line 493 to line 497; page 16, line 518 to line 520; and page 16, line 529 to line 536.

page 15, line 479 to line 480.

The interactive between these TFs and the developmental defects are interesting for future study.

page 16, line 493 to line 597.

We cannot definitively determine the detail pathways in which the newly identified genes participate, nor can we ascertain whether their upregulation is beneficial for embryonic development. Future studies will investigate the detailed signaling pathways involved and the impact on embryonic development through knockdown and overexpression experiments.

page 16, line 518 to line 520.

Previously study found that HIF1-/- MEFs prevent apoptosis under hypoxia because of the reduction of ROS via inhibit the expression of PDK1[12]. This research is contrasted with our research and should be studied in the future.

page 16, line 529 to line 536.

In this study, we investigated the effects of hypoxia (both low oxygen and CoCl2 treatment) on the phenotype and gene expression of mouse embryonic fibroblasts (MEFs) in vitro. We further explored the involvement of the HIF1α signaling pathway in mediating these effects by inhibiting HIF1α activity. Our findings suggest that HIF1α signaling plays a crucial role in regulating MEF phenotype and gene expression under hypoxic conditions. Future studies using gene knockdown and overexpression approaches will be conducted to elucidate the detailed molecular mechanisms by which hypoxia regulates MEF phenotype and gene expression.

Comments on the Quality of English Language

The quality of the English writing should be enhanced.

Response:

Thank you for your feedback regarding the English in the manuscript. We acknowledge the need for improvement and are committed to ensuring that the final version of the manuscript is written in clear, concise, and grammatically correct English.

Reviewer 2 Report

Comments and Suggestions for Authors

Although the present study is valuable, it seems to need substantial revision:
1- The practical purpose of the study is not well explained. Can the results of this study be used for regenerative medicine applications or the treatment of fetal defects?
2- The role of CoCl2 is due to which feature? Why was this combination chosen? Are there similar compounds for this application? If the answer is yes, why was this compound chosen?
3- Due to the number of genes involved, it seems necessary to investigate their interactions via system biology, so discuss the possible interactions between the genes and the activated pathways.
4- The discussion part also needs to compare the current research with other similar cases and compare them.

Author Response

We would like to express our sincere gratitude to all reviewers for their valuable time and insightful comments on our manuscript. We deeply appreciate the constructive feedback you have provided, which has been instrumental in improving the quality of our work. In this point-by-point response letter, reviewer's comments were marked in dark blue italics, followed by our detailed response. All revisions and supplementary new text are clearly highlighted in red.

Comments and Suggestions for Authors

Although the present study is valuable, it seems to need substantial revision:

1- The practical purpose of the study is not well explained. Can the results of this study be used for regenerative medicine applications or the treatment of fetal defects?

Response:

Thank you for your question. We appreciate the reviewer's suggestion and acknowledge the importance of expanding our discussion on the broader implications of our findings. Our results primarily provide a foundation for further research into the mechanisms underlying embryonic development. Additionally, we identify potential targets for therapeutic intervention in developmental defects. We have addressed this point in the revised manuscript by adding a new section to the revised discussion specifically focusing on the potential implications for embryonic development and clinical applications on page 15, line 440 to line 456.

page 15, line 440 to line 456.

The development of a single cell into a complex multicellular organism has been studied for centuries[13]. During this intricate process, any deviation from the precisely orchestrated sequence of events can lead to developmental defects, even resulting in embryonic lethality. Neural tube defects (NTDs) in embryonic development lead to the brain and/or spinal cord[14]. Congenital heart defects (CHDs) can lead to malformation and fetal death[15]. In addition to the diseases mentioned above, other conditions caused by embryonic malformations include cleft lip and palate[16], limb malformations[17], chromosomal abnormalities[18], and single gene disorders[19]. Therefore, understanding the mechanisms underlying this process remains a crucial area of research.

Embryonic fibroblasts are a diverse population of cells that play a vital role in various developmental processes, including tissue morphogenesis and organogenesis. Embryonic fibroblasts promote cardiomyocyte proliferation through the β1 integrin, ERK, and PI3K/Akt pathways and cardiomyocyte-specific β1 integrin KO mice leads to embryonic lethality[20]. These results provide new strategy for heart regenerative therapy. Ski−/− mouse embryo fibroblasts exhibit high levels of genome instability[21]. Therefore, understanding the function of embryonic fibroblasts can contribute to elucidating the mechanisms underlying embryonic malformations and identifying potential therapeutic targets.

Hypoxia plays an crucial role in embryonic development, such as organogenesis, cell differentiation, and vascularization[22]. Mammalian embryos develop under hypoxic conditions, previous study have shown that the deletion of HIF1a, a key regulator of the hypoxia response, leads to embryonic lethality[23]. This suggests the critical role of the hypoxia signaling pathway in embryonic development. Although hypoxia is essential for embryonic development, and fibroblasts play a vital role in embryonic development, the specific changes that occur in embryonic fibroblasts under hypoxic remain unclear. Here, we demonstrate that hypoxia can induce the multiple changes of MEFs and stimulated multiple signaling pathways.

2- The role of CoCl2 is due to which feature? Why was this combination chosen? Are there similar compounds for this application? If the answer is yes, why was this compound chosen?

Response:

Thank you for your question. We appreciate the opportunity to clarify the role of CoCl2 in our study. CoCl2 functions by mimicking the effects of hypoxia. It achieves this by replacing Fe2+ in the oxygen-sensing heme group of the hypoxia-inducible factor (HIF) prolyl hydroxylase (PHD) enzyme. This replacement inhibits PHD activity, preventing the degradation of HIF-1α, a key regulator of the cellular response to low oxygen levels. Consequently, HIF-1α accumulates, leading to the expression of genes associated with hypoxic adaptation, including those involved in angiogenesis, glycolysis, and cell survival.

CoCl2 is a well-established and widely used chemical inducer of HIF-1. It has been extensively studied and validated in numerous research publications, providing a strong foundation for our study; CoCl2 is readily available and affordable, making it a practical choice for research applications. Additionally, its simple preparation and administration procedures facilitate experimental design and execution.

While CoCl2 is a widely used HIF-1 inducer, other compounds with similar mechanisms of action exist, including desferrioxamine (DFO)[24], dimethyloxalylglycine (DMOG)[25]. However, CoCl2 was chosen for this study due to its well-established reputation, affordability, ease of use, and extensive literature support. We have included a description of CoCl2 in our revised manuscript from page 2, line 74 to line 81.

page 2, line 74 to line 81.

CoCl2 is an iron chelator which functions by mimicking the effects of hypoxia[24]. It achieves this by replacing Fe2+ in the oxygen-sensing heme group of the hypoxia-inducible factor (HIF) prolyl hydroxylase (PHD) enzyme. This replacement inhibits PHD activity, preventing the degradation of HIF-1α, a key regulator of the cellular response to low oxygen levels[26]. Consequently, HIF1α accumulates, leading to the expression of genes associated with hypoxic adaptation, including those involved in angiogenesis, glycolysis, and cell survival[27]. CoCl2 is widely used to mimic hypoxic conditions in a variety of cell types, such as HepG2[28], MCF-7 cells[29], and colorectal cancer cells[30].

3- Due to the number of genes involved, it seems necessary to investigate their interactions via system biology, so discuss the possible interactions between the genes and the activated pathways.

Response:

We appreciate the reviewer's suggestion to discuss the possible interactions between the genes identified in our study and the activated pathways. We agree that this is an important aspect to consider, and we have added the possible interactions between the genes and the activated pathways on page 11, line 339 to line 353.

page 11, line 339 to line 353.

These genes have not been traditionally considered to be involved in the hypoxic response. Proser 2 were reported to suppresses invasion by increasing the level of p-AMPK in pancreatic ductal adenocarcinoma (PDAC)[31]. Thus, Proser 2 may be involved in the migration of MEFs under hypoxia. Bean1 can interact with NEDD4, which is developmentally regulated, and is highly expressed in embryonic tissues [32, 33]. Bean1 is upregulated under hypoxia, this suggested that Bean1 may be associated with embryonic development. Mutations in Bean1 are associated with spinocerebellar ataxia type31[34]. Rnf128 is an E3 ubiquitin ligase, which involved in multiple disease, such as acute lung injury[35], hepatocellular carcinoma[36], and colorectal cancer(CRC)[37]. Rnf128 also participated in regulating CRC via PI3K-Akt signaling pathway[37]. The upregulation of Rnf128 under hypoxia in MEFs may participated in regulating the embryonic development through PI3K-Akt signaling pathway. Dpf1 was predicted to be involved in negative regulation of transcription, nervous system development, and positive regulation of transcription by RNA polymerase II. The function of Fam71f1 and Dpf1 have not been studied.

4- The discussion part also needs to compare the current research with other similar cases and compare them.

Response:

We appreciate the reviewer's suggestion and acknowledge the importance of compare previous research with ours’ study. We have compared the TFs that response to hypoxia on page 15, line 466 to line 478. We have now added the comparison in the revised manuscript on page 16, line 482 to line 484, page 16, line 517 to line 519.

Page 16, line 482 to line 484.

Our results indicated that HIF1α is activated under hypoxia (Figure 1, 3A), these results are consisted with the study of Randall’s group[38].

Page 16, line 518 to line 520.

Previously study found that HIF1-/- MEFs prevent apoptosis under hypoxia because of the reduction of ROS via inhibit the expression of PDK1[12]. This research is contrasted with our research and should be studied in the future.

Reviewer 3 Report

Comments and Suggestions for Authors

This study looks at the hypoxic processes in fibroblasts with respect to ROS production and apoptosis. I have some points as below:

1. 1C is too low resolution to judge the differences especially as the difference appears to be minimal as seen in 1D. Is this biologically relevant?

2. The upregulated genes are interestingly all involved in similar processes. Do teh authors think this helps hypoxia? Are there other studies that support upregulation of these same processes in hypoxic conditions?

3. Is there a reason why the Hif1 does not affect Ldha and Glut1 as much as Pdk1? The difference between the 3 readouts is striking.

4. Again, 3G,O,M are not clear images and cannot be independently evaluated. Especially as they are quantified. Images need to be higher resolution and require insets with arrows pointing to what the reader is supposed to see.

5. BY novel HIf1 mediated gene upregulation, do the authors mean they have not been reported earlier? Has such studies been done? I am not sure what novel means here. Are these genes unexpected in hypoxic response? What might their functions be in hypoxia?

6. Are the datasets uploaded to a database? That should be done before publication. Or a full list of up and down regulated genes with the corresponding scores should be provided. This is the international standard for any large datasets, especially ones with KEGG analysis. It is quite nice that the authors validated some of the targets but without reporting the dataset fully, the readers cannot build on this study, nor can the results be evaluated or validated independently.

Minor points:

In several places, the English language editing makes it hard to read. For e.g. in certain places "the migration..." should not have an article. "Virify" instead of "verify" and other grammar errors should be corrected.

Comments on the Quality of English Language

There should be some editing done. Please proof the English and spelling carefully. Please see my minor comments above. 

Author Response

We would like to express our sincere gratitude to all reviewers for their valuable time and insightful comments on our manuscript. We deeply appreciate the constructive feedback you have provided, which has been instrumental in improving the quality of our work. In this point-by-point response letter, reviewer's comments were marked in dark blue italics, followed by our detailed response. All revisions and supplementary new text are clearly highlighted in red.

Comments and Suggestions for Authors

This study looks at the hypoxic processes in fibroblasts with respect to ROS production and apoptosis. I have some points as below:

1.1C is too low resolution to judge the differences especially as the difference appears to be minimal as seen in 1D. Is this biologically relevant?

Response:

Thank you for your feedback. We have updated revised Figure 1C to improve the image resolution. We believe that the observed increase in MEFs migration in response to hypoxia is biologically relevant. Hypoxia is known to induce the expression of a variety of genes that promote cell migration, including vascular endothelial growth factor (VEGF) and hypoxia-inducible factor 1 (HIF1α). These genes play important roles in angiogenesis and migration, which are processes that are often associated with hypoxia. Our findings are consistent with previous studies that have shown that hypoxia can induce cell migration in a variety of cell types, including endothelial cells[39], fibroblasts[40], and cancer cells[41, 42].

2.The upregulated genes are interestingly all involved in similar processes. Do the authors think this helps hypoxia? Are there other studies that support upregulation of these same processes in hypoxic conditions?

Response:

Thank you for your insightful observation. We do believe that the upregulation of these genes is likely to be beneficial for cells under hypoxic conditions. The upregulated genes are all involved in processes that are essential for cell survival and adaptation to hypoxia. For example, the genes involved in glycolysis and lactate production (Ldha, Glut1, and Pdk1) allow cells to generate energy in the absence of oxygen[43]. The genes involved in angiogenesis (Vegfa) promote the formation of new blood vessels, which can improve oxygen delivery to the tissues[43]. We cannot definitively determine the detail pathways in which the newly identified genes participate, nor can we ascertain whether their upregulation is beneficial for embryonic development. Future studies will investigate the detailed signaling pathways involved and the impact on embryonic development through knockdown and overexpression experiments. We have added these to the revised discussion on page 16, line 493 to line 497.

page 16, line 493 to line 497.

However, we cannot definitively determine the detail pathways in which the newly identified genes participate, nor can we ascertain whether their upregulation is beneficial for embryonic development. Future studies will investigate the detailed signaling pathways involved and the impact on embryonic development through knockdown and overexpression experiments.

  1. Is there a reason why the Hif1 does not affect Ldha and Glut1 as much as Pdk1? The difference between the 3 readouts is striking.

Response:

Thank you for your question. The differential effects of CoCl2 on Ldha, Glut1, and Pdk1 are likely due to off-target effects of CoCl2, CoCl2 is a chemical inducer that can lead to the accumulation of HIF1, which in turn can upregulate the expression of HIF1 downstream target genes. However, it is important to note that chemical agents can also trigger other unknown reactions. Therefore, it is possible that the expression patterns of Pdk1 induced by CoCl2 may differ from those of Ldha and Glut1. The specific mechanisms underlying these differences require further investigation in future studies. We have added these to the revised discussion on page 16, line 505 to line 511.

page 16, line 505 to line 511.

The differential effects of CoCl2 on Ldha, Glut1, and Pdk1 are likely due to off-target effects of CoCl2. CoCl2 is a chemical used to mimic low oxygen conditions, can affect the expression of genes involved in energy metabolism in embryonic fibroblasts. However, it is important to note that CoCl2 can also have other effects on cells, which may explain why it affects the expression of some genes differently than others. More research is needed to understand the exact mechanisms behind these differences.

  1. 4. Again, 3G, O, M are not clear images and cannot be independently evaluated. Especially as they are quantified. Images need to be higher resolution and require insets with arrows pointing to what the reader is supposed to see.

Response:

Thank you for bringing this to our attention. We apologize for the lack of clarity in Figures 3G, O, and M. We have carefully reviewed the figures and have made the following changes:

We have replaced the original low-resolution images with high-resolution versions in revised Figure 3. The new images are significantly clearer and provide a more detailed view of the experimental results.

We have added insets to Figures 3G and O with white arrows pointing to the specific features that are being quantified. This will make it easier for readers to understand the results and to independently evaluate the images. Revised Figure 3 Legends: We have updated the figure legends to provide more detailed descriptions of the specific features that are being highlighted in the insets on page 11, line 322 to line 323.

page 11, line 322 to line 323.

The white arrow in G and O represents cells that are producing ROS cells.

  1. 5. BY novel HIf1 mediated gene upregulation, do the authors mean they have not been reported earlier? Has such studies been done? I am not sure what novel means here. Are these genes unexpected in hypoxic response? What might their functions be in hypoxia?

Response:

Thank you for your question. We apologize for the lack of clarity in our terminology. By "novel HIF1α-mediated gene upregulation," we meant that these genes have not been previously reported to be involved in the hypoxic response. We have conducted a thorough literature search to identify previous studies that have investigated the regulation of these genes by HIF1α. We have found that some of these genes have been reported to be upregulated by HIF1α in other studies. For example, Ldha, Glut1, and Pdk1 are well-established HIF1α target genes. Ldha, Pdk1, and Glut1 are directly involved in energy metabolism[43]. Vegfa are involved in angiogenesis[43]. However, some genes have not been traditionally considered to be involved in the hypoxic response, such as our list genes. We have included a description of the new genes in our revised manuscript from page 11, line 339 to line 353.

page 11, line 339 to line 353.

These genes have not been traditionally considered to be involved in the hypoxic response. Proser 2 were reported to suppresses invasion by increasing the level of p-AMPK in pancreatic ductal adenocarcinoma (PDAC)[31]. Thus, Proser 2 may be involved in the migration of MEFs under hypoxia. Bean1 can interact with NEDD4, which is developmentally regulated, and is highly expressed in embryonic tissues [32, 33]. Bean1 is upregulated under hypoxia, this suggested that Bean1 may be associated with embryonic development. Mutations in Bean1 are associated with spinocerebellar ataxia type31[34]. Rnf128 is an E3 ubiquitin ligase, which involved in multiple disease, such as acute lung injury[35], hepatocellular carcinoma[36], and colorectal cancer(CRC)[37]. Rnf128 also participated in regulating CRC via PI3K-Akt signaling pathway[37]. The upregulation of Rnf128 under hypoxia in MEFs may participated in regulating the embryonic development through PI3K-Akt signaling pathway. Dpf1 was predicted to be involved in negative regulation of transcription, nervous system development, and positive regulation of transcription by RNA polymerase II. The function of Fam71f1 and Dpf1 have not been studied.

  1. Are the datasets uploaded to a database? That should be done before publication. Or a full list of up and down regulated genes with the corresponding scores should be provided. This is the international standard for any large datasets, especially ones with KEGG analysis. It is quite nice that the authors validated some of the targets but without reporting the dataset fully, the readers cannot build on this study, nor can the results be evaluated or validated independently.

Response:

Thank you for your question and for highlighting the importance of data availability. We apologize for not including the accession number for our transcriptomic data in the original manuscript. We have now uploaded our data to the NCBI Gene Expression Omnibus (GEO) database and have included the accession number in the Materials and Methods section of the revised manuscript on page 5, line 190 to line 191.

In addition to providing the accession number, we have also included a supplementary table (Table S1) that lists all of the differentially expressed genes identified in our analysis, along with their corresponding fold changes and p-values. This table will allow readers to easily access and analyze the full dataset.

page 5, line 190 to line 191.

The raw data from our RNA-sequencing experiments have been deposited in the GEO database (accession number: PRJNA1134336).

Minor points:

In several places, the English language editing makes it hard to read. For e.g. in certain places "the migration..." should not have an article. "Virify" instead of "verify" and other grammar errors should be corrected.

Response:

     Thank you for bringing this to our attention. We apologize for the errors in language editing and understand that they can make the manuscript difficult to read. We have carefully reviewed the manuscript and have made the following changes:

We have removed the article "the" in cases where it was not needed, such as "migration of cells" instead of "the migration of cells."

We have corrected the spelling of " virify " to " verify " on page 12, line 355, and any other spelling errors that we found. We have made minor grammatical corrections throughout the manuscript.

Comments on the Quality of English Language

There should be some editing done. Please proof the English and spelling carefully. Please see my minor comments above. 

Response:

Thank you for your feedback regarding the English in the manuscript. We acknowledge the need for improvement and are committed to ensuring that the final version of the manuscript is written in clear, concise, and grammatically correct English.

Reviewer 4 Report

Comments and Suggestions for Authors

While the study offers valuable insights into hypoxia-induced adaptations in embryonic fibroblasts, it falls short in elucidating the precise mechanisms by which hypoxia and CoCl2 differentially regulate these processes. The findings would benefit from further experiments, such as gene knockdown or overexpression studies, to provide deeper mechanistic understanding. Additionally, the manuscript lacks a thorough discussion on the broader implications of these findings for embryonic development and potential clinical applications, which would enhance the reader's appreciation of the study's significance. The statistical analysis methods are also not consistently detailed, and ensuring clarity and consistency in reporting these methods would strengthen the overall robustness of the study.

Author Response

We would like to express our sincere gratitude to all reviewers for their valuable time and insightful comments on our manuscript. We deeply appreciate the constructive feedback you have provided, which has been instrumental in improving the quality of our work. In this point-by-point response letter, reviewer's comments were marked in dark blue italics, followed by our detailed response. All revisions and supplementary new text are clearly highlighted in red.

Comments and Suggestions for Authors

While the study offers valuable insights into hypoxia-induced adaptations in embryonic fibroblasts, it falls short in elucidating the precise mechanisms by which hypoxia and CoCl2 differentially regulate these processes. The findings would benefit from further experiments, such as gene knockdown or overexpression studies, to provide deeper mechanistic understanding. Additionally, the manuscript lacks a thorough discussion on the broader implications of these findings for embryonic development and potential clinical applications, which would enhance the reader's appreciation of the study's significance. The statistical analysis methods are also not consistently detailed, and ensuring clarity and consistency in reporting these methods would strengthen the overall robustness of the study.

Response:

Thank you for your question. We acknowledge that our research represents a preliminary investigation into the effects of hypoxia and CoCl2 on embryonic fibroblasts. We have addressed this point in the revised manuscript as follows: We have explicitly stated in the revised discussion that our study is a preliminary exploration of the topic. We have emphasized the need for further research to confirm and expand upon our observations. We have discussed the limitations of our study and the need for further research. We have included a description of CoCl2 in our revised manuscript from page 16, line 529 to line 536.

We also appreciate the reviewer's suggestion and acknowledge the importance of expanding our discussion on the broader implications of our findings. We have addressed this point in the revised manuscript by adding a new section to the revised discussion specifically focusing on the potential implications for embryonic development and clinical applications on page 15, line 440 to line 456.

We appreciate the reviewer's feedback and acknowledge the importance of providing clear and consistent details about our statistical analysis methods. We have re-analyzed our experimental data and have added detailed statistical methods to the revised materials and methods section on page 5, line 194 to line 203.

page 16, line 529 to line 536.

In this study, we investigated the effects of hypoxia (both low oxygen and CoCl2 treatment) on the phenotype and gene expression of mouse embryonic fibroblasts (MEFs) in vitro. We further explored the involvement of the HIF1α signaling pathway in mediating these effects by inhibiting HIF1α activity. Our findings suggest that HIF1α signaling plays a crucial role in regulating MEF phenotype and gene expression under hypoxic conditions. Future studies using gene knockdown and overexpression approaches will be conducted to elucidate the detailed molecular mechanisms by which hypoxia regulates MEF phenotype and gene expression.

page 15, line 440 to line 456.

The development of a single cell into a complex multicellular organism has been studied for centuries[13]. During this intricate process, any deviation from the precisely orchestrated sequence of events can lead to developmental defects, even resulting in embryonic lethality. Neural tube defects (NTDs) in embryonic development lead to the brain and/or spinal cord[14]. Congenital heart defects (CHDs) can lead to malformation and fetal death[15]. In addition to the diseases mentioned above, other conditions caused by embryonic malformations include cleft lip and palate[16], limb malformations[17], chromosomal abnormalities[18], and single gene disorders[19]. Therefore, understanding the mechanisms underlying this process remains a crucial area of research.

Embryonic fibroblasts are a diverse population of cells that play a vital role in various developmental processes, including tissue morphogenesis and organogenesis. Embryonic fibroblasts promote cardiomyocyte proliferation through the β1 integrin, ERK, and PI3K/Akt pathways and cardiomyocyte-specific β1 integrin KO mice leads to embryonic lethality[20]. These results provide new strategy for heart regenerative therapy. Ski−/− mouse embryo fibroblasts exhibit high levels of genome instability[21]. Therefore, understanding the function of embryonic fibroblasts can contribute to elucidating the mechanisms underlying embryonic malformations and identifying potential therapeutic targets.

Hypoxia plays an crucial role in embryonic development, such as organogenesis, cell differentiation, and vascularization[22]. Mammalian embryos develop under hypoxic conditions, previous study have shown that the deletion of HIF1a, a key regulator of the hypoxia response, leads to embryonic lethality[23]. This suggests the critical role of the hypoxia signaling pathway in embryonic development. Although hypoxia is essential for embryonic development, and fibroblasts play a vital role in embryonic development, the specific changes that occur in embryonic fibroblasts under hypoxic remain unclear. Here, we demonstrate that hypoxia can induce the multiple changes of MEFs and stimulated multiple signaling pathways.

page 5, line 194 to line 203.

Data normality and homogeneity of variance were assessed using the Shapiro-Wilk. Two-group comparisons employed t-tests for equal variances and normal distributions, Wilcoxon tests for equal variances and non-normal distributions, and Welch's t-tests for unequal variances and normal distributions. For more than two independent groups, one-way ANOVAs with Tukey's post hoc tests were used for equal variances and normal distributions, while Kruskal-Wallis tests with Dunn's post hoc tests were employed for equal variances and non-normal distributions. Welch's ANOVAs with Games-Howell post hoc tests were applied for unequal variances and normal distributions. A significance level of P < 0.05 was adopted for all analyses. All statistical procedures were conducted using Prism 10.

Round 2

Reviewer 1 Report

Comments and Suggestions for Authors

The revised manuscript looks good to me. It can be accepted for publication.

Reviewer 2 Report

Comments and Suggestions for Authors

The manuscript has been revised appropriately and can be considered for publication.